# Effect of Ploughing Techniques on Water Use and Yield of Rice in Maugo Small-Holder Irrigation Scheme, Kenya

Pius Kipchumba Cheboi [1], Shahida Anusha Siddiqui [2,3],*, Japheth Onyando [4], Clement Kiprotich Kiptum [5] and Volker Heinz [3]

1 Department of Agricultural and Biosystems Engineering, School of Engineering, University of Eldoret, Eldoret 1125-30100, Kenya; picheboi2014@gmail.com

2 Department of Biotechnology and Sustainability, Technical University of Munich (TUM), 94315 Straubing, Germany

3 DIL e.V.–German Institute of Food Technologies, 49610 D-Quakenbrück, Germany; v.heinz@dil-ev.de

4 Department of Agricultural and Biosystems Engineering, Faculty of Engineerng and Technology, Egerton University, Egerton-Njoro 536-20115, Kenya; jonyando@gmail.com

5 Department of Civil and Structural Engineering, School of Engineering, University of Eldoret, Eldoret 1125-30100, Kenya; ckiptum@uoeld.ac.ke

* Correspondence: s.siddiqui@dil-ev.de

**Abstract:** The objective of this study was to determine the effect of paddy rice ploughing techniques on water use and the yield of rice crop, as well as water use efficiency for rice growing in small-holder irrigation schemes. The study was conducted at a farmer's field in Powo B sub-block of Maugo Irrigation Scheme. The period of study was from July 2019 to January 2020, which is the rice season. The experimental site was located in the vicinity of Olare Shopping Centre, Kamenya Sub-location, Kochia East Location, Kochia Ward, Rangwe Sub-County, Homa Bay County, Nyanza Region, Kenya in Maugo rice scheme in Kenya. In the study, four irrigation tillage practices were applied: ox-plough, conventional ox-plough, hand hoe and tractor ploughing. The results showed that conventional ox-ploughing consumed the highest amount of water at 1240 mm. The highest water use efficiency of 0.49 kg/m$^3$ and highest yield of 5.7 tons/ha were observed for hand hoe ploughing. Use of the hand hoe ploughing technique increased yields by 20 percent, as compared to the conventional ox-ploughing. Therefore, the use of water for ploughing is not necessary in the study area. Future research will be needed to see how farmers are adopting the technology before scaling up to full mechanization, as partial mechanization was not profitable.

**Keywords:** hand hoe; ox plough; ploughing techniques; water use; yield

## 1. Introduction

Rice from Asia was first introduced in Kenya in 1907 and is the third most important cereal crop after maize and wheat. Despite the annual increase in rice consumption at a rate of 1% in Kenya, production of rice has not managed to keep up with consumption as it has been fluctuating in the past 20 years between 45,000 and 80,000 tons per year. This forces Kenya to import rice annually because the national rice consumption is estimated to be 300,000 tons [1]. Rice production in Kenya between 2014–2018 stood at 3.84 tons/ha [2].

In Kenya, 80% of rice is grown in irrigation schemes. There are more than 3000 existing small-holder irrigation schemes in Mwea, Bunyala, Ahero, Kano, Msabweni, Maugo and Tana deltas among other areas where rice is grown for food and commercial purposes [1]. This shows that small-holder farmers are vital for Kenya's agriculture and rural economy, as is the case in India [3]. Small-holder farmers in Kenya farm smaller pieces of land, with a sizes 0.47 hectares [4]. Maugo Irrigation Scheme is managed by Maugo Rice Farmers Co-operative Society. The scheme is a gravity-operated scheme and is located in Homa Bay county, Kenya. It was started by farmer's initiatives in the year 1962 using flood water

from River Maugo for subsistence. Land sizes in the scheme are small and measure 40 m by 100 m and are ploughed using different techniques [1].

Small-holder farmers use hand held hoe, ox and tractor for ploughing. These ploughing techniques have their own unique challenges. All families own two to three hoes; however, during ploughing when more hoes are needed, some homesteads borrow these from their neighbors. Borrowing of hoes can delay ploughing if neighbors are using them at the same time. Oxen for ploughing can be owned by the farmer or hired from another farmer. During ploughing, the demand for oxen goes up, while some farmers are left behind. Many tractors ploughing fields are mainly hired from the rich farmers from the neighborhood, but they are expensive [1] and out of reach for poor farmers. The majority of these tractors are old and inefficient, thus resulting in the emission of greenhouse gases (GHGs) through the consumption of fossil fuels [5]. All the challenges end up delaying ploughing and thereby affecting crop production. Poor and untimely land preparation cause serious weed problems and expose plants to harmful substances such as carbon dioxide and butyric acid released by decaying organic matter in the soil [6]. When not done properly, ploughing results in unevenness of the paddy field, resulting in the uneven growth of rice and thus reduced yield.

Depth of ploughing depends on the type of tool used. Hand hoe ploughing depths are 10 cm and can be repeated to achieve deeper depths. Animal drawn ploughs, like ox-plough, plough to a depth of 15 cm. Disc plough attached to tractors can plough to greater depths, such as 30 cm [7]. Recommended depths of ploughing for rice are 15–20 cm. Deep ploughing should be discouraged because deeper ploughing moves the fertile soil deeper and hence unavailable for rice growth [8]. Ploughing is followed by harrowing and then levelling before transplanting seedlings. Ploughing and harrowing in water is known as puddling. Puddling makes the soil not lose water by deep percolation. Experiments on rice have shown that tillage has greater effects on yields than irrigation. Ploughing to recommended depth can be one way in which the Kenyan government can increase annual production of rice from 70,000 tons in 2018 to 406, 456 tons by the year 2022 [9]. Tillage to depths of 20–25 cm favors root development and uptake of minerals and hence higher yields of 5.82 tons/hectare could be achieved [10].

Farmers in Maugo Rice Irrigation Scheme barely subsist as they are stuck in the poverty cycle. Most of these farmers cannot access credit from commercial banks as the majority of them (48%) do not have title deeds for their farms [11]. Hence, these farmers, like other small-holder farmers, face myriad of challenges like human capital, infrastructure for ploughing, irrigation water [12], among other problems like climate change. This calls for a study on water management at the farmers level with a view to saving water, which is a priority for any farmer. As climate change is expected to reduce rice yields by 10% in the 21st century [13], farmers need to be taught methods of saving water for them to be able to cope with water challenges in the future. Farmers in Maugo area do not plough dry soil, rather they add some water to make it soft. Addition of this water helps in ploughing, but will not contribute to crop development. Ploughing dry soil helps to save water [14], but results in decreased yields. Saving of water will address the water problem particularly in areas where canals are not working effectively and low water flows in rivers that are normally observed in the study area during dry months [11]. This will further give information on the performance of dry-seeded rice [15]. Some literature [16] has focused on the system of rice intensification on large scale and not small-holder farmers, which are key to rice production in Kenya. Understanding local practices done by small scale farmers has generated plenty of research interest and thus requires attention and further scrutiny. Pesticide application practices and knowledge of small-scale farmers was done by Ndayambaje et al. 2019 [17] in Rwanda.

Land preparation techniques may influence water retention and yields of rice. Zero-tillage is considered to be the best alternative to ploughing and harrowing due to its effect on soil properties, but ploughing, harrowing and levelling increases rice yield and reduces weed density. Indeed, most developing countries like Kenya have not benefitted from

the awareness of zero-tillage [18]. Information of the effect of different land preparation techniques on water retention and rice yields in smallholder irrigation schemes like Maugo Irrigation scheme is scanty. While many studies have proposed ways of improving water management, little is known about the performance of smallholder farmers' ploughing technique practices and other ploughing practices in reducing water use in rice growing. This research highlights the need for farmers to slowly learn new ways to improve their technical know-how on water management by setting up experiments near them.

The objective of this study was to compare water retention and rice yield at a farmer's plot using conventional tillage of wetting before ploughing and ploughing without wetting using hand hoe, oxen and tractor as ploughing techniques.

## 2. Materials and Methods

### 2.1. Study Area

This study was conducted on a farmer's field in Powo B sub-block of Maugo Irrigation Scheme during July 2019–January 2020 rice cropping season. The experimental site was located in the vicinity of Olare Shopping Centre, Kamenya Sub-location, Kochia East Location, Kochia Ward, Rangwe Sub-County, Homa Bay County, Nyanza Region, Kenya at geographical co-ordinates 0°30′ S and 34°30′ E (Figure 1). The Population of the study area is 10,193 [19].

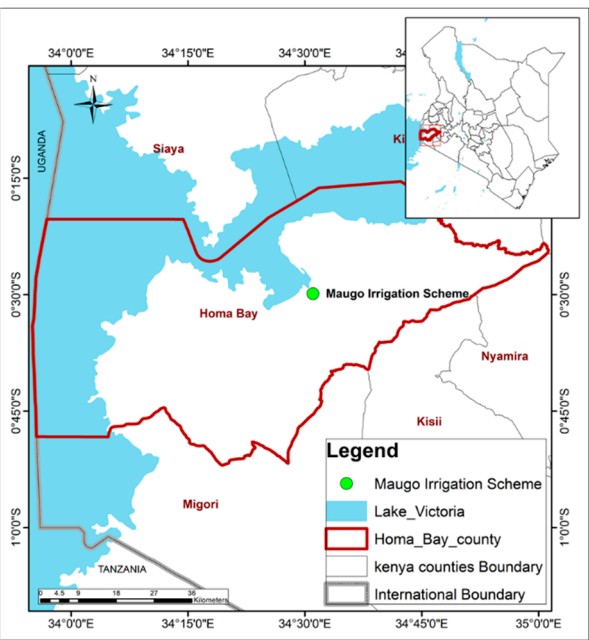

**Figure 1.** Map of study area showing the Maugo irrigation scheme.

The study area altitude varies from 1145 to 1190 m above sea level. Agro-ecologically, the area is sub-humid, lower midland (LM3), suitable for growing maize, sorghum, cow peas, ground nuts, beans, soya, sweet potatoes, sunflower, sesame, green grams, rice and vegetables. The mean annual rainfall ranges from 800 mm to 1200 mm with long season whose peak is between April and May and season that peaks between November and December. Annual mean maximum and minimum temperatures is 31 °C and 18 °C, respectively. Relative humidity varies between 60% and 75% and potential evapo-transpiration of 1744 mm/annum. It has an altitude of 1145–1190 m above sea level. The area is located on the lowland alluvial plains. The soil texture is clay (vertisols). The area is generally fertile as they originate from nutrient rich alluvial deposits washed downstream [14].

The source of water for irrigation in the scheme is Maugo River, which is a seasonal tributary of River Tende with a base flow of 0.5 m³/s and its catchment extends to Kisii hills,

the tail end of the catchment area of 1700 km$^2$, with a flood return period of 5 years carrying 79–90 m$^3$/s. The area is a seasonal swamp prone to flooding during heavy rains [14].

### 2.2. Land Preparation Techniques

The following combinations of land preparation techniques were applied:

1. Tractor ploughing followed by hand harrowing then hand levelling.
2. Ox ploughing then hand harrowing then hand levelling.
3. Hand ploughing then hand harrowing then hand levelling.
4. Conventional ox ploughing then hand harrowing followed by hand levelling.

### 2.3. Experimental Design

The layout of the experiment was a Complete Randomized Block Design (CRBD) with four ploughing techniques and four replicates, as shown in Table 1. A total of 16 plots of 10 m * 9 m each (90 m$^2$) were measured using a tape measure with a total area for the research farm as 1440 m$^2$ (0.144 ha). The plots were pegged for easy identification of the boundaries of the plots during ploughing and random allocation of treatments to them.

**Table 1.** Design layout of ploughing treatments.

| Replications in Plots | | | |
|---|---|---|---|
| **1** | **2** | **3** | **4** |
| Conventional | Tractor | Hand | Ox |
| Ox | Conventional | Tractor | Hand |
| Hand | Ox | Conventional | Tractor |
| Tractor | Hand | Ox | Conventional |

### 2.4. Seperation of Plots

Each plot was ploughed using the identified land preparation techniques, separated and guarded by using mud bunds. Inlets and outlets were made for the measurement of water inflows and outflows. Bunds of 0.25 m high and 0.25 m wide were made by heaping soil to separate plots. Bunds destroyed by floods were quickly sealed. The bunds to control the water and separate the plots were made before ploughing and those for harrowing were made later. Earthen water inlets and outlets were made to facilitate the measurement of water inflows and outflows.

### 2.5. Rice Variety

Hybrid rice seed Arize 6444 Gold from Bayer Company was selected based on high yield potential, resistance to shattering and diseases, high milling yield, good eating qualities and suitability for the market. High-quality seed reduces the required seeding rate and produces strong, healthy seedlings, resulting in a more uniform crop with higher yields. The seeds were soaked in water for 24 h to break dormancy and then removed and spread in an open area to sprout. The sizes of seedbeds were 28 m$^2$ in these areas. They were ready in three weeks (about 21 days plus one day) for transplanting.

### 2.6. Crop Management at Nursery and in the Field

A nursery site was selected and prepared by ploughing and harrowing twice. Each experimental plot was harrowed and leveled under water by hand hoe to allow uniform water ponding. The plots were harrowed twice at an interval of three days to ensure proper soil-water mixture [15]. Transplanting was done in lines into puddled and water-covered fields after 20–30 days of germination in all the treatments on 5 and 6 September 2019. Fertilizer was broadcasted, then left for 2–3 h to dissolve before transplanting. Row planting was used (0.15 m by 0.20 m). Fertilizer was broadcasted at a rate of 50 kg per acre for Di-ammonium phosphate (DAP). Top dressing at the rate of 50 kg per acre was done using Sulphate of Ammonia (SA). Weeding was done within the first 20–50 days after

crop establishment and the field was maintained weed free throughout the growing season. Harvesting was carried out using sickles and put in tarpaulins the same day and threshed immediately in the field. This was done to avoid theft and destruction by floods at night if left in the field. Water was drained 1–2 weeks before harvesting. Threshing was done by manual labor. Straw was left in the farm as animal feed. Grains were dried for two sunny days for all the treatments, as practiced by farmers and stored for 2–3 months before milling to determine the yield.

### 2.7. Data Collection and Analysis

The depth of water in millimeters (mm) entering and leaving each plot was measured weekly for determination of the amount of water retained. During crop growth, rice tillers at 14th day and 77th day after transplanting were counted individually and recorded. Rooting depth was measured using a transparent one-meter ruler after harvesting. This was achieved by digging the ground to expose the roots. After harvesting the grains from each plot, they were measured and recorded for comparison purposes. Data on yield and water used was used to calculate Water Use Efficiency (WUE). Yield divided by water used resulted in WUE. Data was analyzed in Microsoft Excel®. Statistical significance was done at $\alpha$ = 0.05 on the basis of F tests that is normally used to compare variances. Separation of means was done using Fisher's list significant difference (LSD)-tests at $\alpha$ = 0.05.

## 3. Results

### 3.1. Effect of Ploughing Technique on Rooting Depth and Number of Tillers

Conventional tillage and ox plough had the minimum number of tillers of 19 after two weeks of transplanting (Table 2). Hand hoe ploughing had the highest number of tillers of 23 tillers. Tractor ploughing had the second highest number of tillers at week two after transplanting. On the 77th day (11 weeks) after transplanting, conventional and ox plough still had the lowest number of tillers. The highest number of tillers were observed for both hand and tractor ploughed plots. There was a slow increase in the number of tillers between day 14 and day 77 for hand ploughed plots when compared with the other treatments. There was no significant difference in the number of tillers for the different treatments. Maximum rooting depth of 51 cm at harvest were in tractor ploughed with hand hoe ploughing coming in second with a depth of 48 cm. The lowest rooting depths of 46 cm were observed for conventional and ox ploughing. However, the rooting depths were significantly different for the treatments. There were differences between tractor ploughing and all the other treatments. Further, there were differences in rooting depth between ox-ploughing and hand hoe ploughing. This was because the Fcritical of 3.49 was lower than F calculated. In addition, their Fisher's LSD were less than the absolute means.

**Table 2.** Mean number of tillers and rooting depth for different ploughing techniques.

| Description | Ploughing Treatments | | | | Fcalculated |
|---|---|---|---|---|---|
| | Conventional | Ox | Hand | Tractor | |
| 14-day tillers (number) | 19 | 19 | 23 | 21 | 0.85 |
| 77-day tillers (number) | 23 | 23 | 24 | 24 | 0.85 |
| Rooting depth (cm) | 46 | 46 | 48 | 51 | 10.37 |

### 3.2. Effect of Ploughing Technique on Amount of Water Retained, Yield and Water Use Efficiency

The amount of water used in preparation of the farmers' field (conventional) in the scheme was 380 mm. This value is the sum of water used during ploughing, harrowing and levelling (Table 3). Conventional ox-ploughing consumed the highest amount of water (1240 mm). Tractor ploughed fields had the second highest amount of water consumed. The least amount of water used was observed in ox-ploughed fields. Paddy rice field prepared using hand hoe ploughing resulted to a significantly higher mean milled grain weight of 5.7 tons/ha followed by a paddy rice field prepared using tractor ploughing of 5.6 tons/ha followed by third ox-ploughing of 5.1 tons/ha. The yields were significantly

different for all the treatments because the F calculated was more than critical F of 3.49, except yields between ox-plough and hand hoe ploughing were not significantly different. This was because LSD was less than the absolute mean of the differences. The water use efficiency (WUE) ranged between 0.38 and 0.49 kg/m$^3$ with the highest value achieved by hand hoe ploughing.

**Table 3.** Mean water used and rice yield for different ploughing techniques.

| Description | Ploughing Treatments | | | |
|---|---|---|---|---|
| | Conventional | Ox | Hand | Tractor |
| Ploughing (mm) | 200 | Nil | Nil | Nil |
| Harrowing (mm) | 110 | 120 | 130 | 130 |
| Levelling (mm) | 70 | 80 | 70 | 80 |
| Crop development (mm) | 860 | 890 | 970 | 990 |
| Total water used (mm) | 1240 | 1090 | 1170 | 1200 |
| Yield (tons/ha) | 4.7 | 5.1 | 5.7 | 5.6 |
| WUE (kg/m$^3$) | 0.38 | 0.47 | 0.49 | 0.47 |

## 4. Discussion

Tractor ploughed plots had the highest number of tillers owing to its greater depths. The slow increase of tillers between week 2 and week 11 for hand ploughed fields could be attributed to shallow depths for hand hoe ploughing. The maximum number of tillers observed in day 77 was consistent with modern day rice varieties that have 20–25 tillers [20]. Tractor ploughing had highest depth due to the increasing intensity of tillage. This means that, with greater depths, came adequate aeration for the growth of rice. Increase in root volume means an increase in nutrient uptake by the root, which led to high yields. All the rooting depths were higher than 45 cm observed for well-watered rice varieties in Malaysia [21] and more than double of what was observed in Taiwan [22].

The amount of water used during land preparation was close to 360 mm observed by Singh, et al. 2001 [23]. The highest consumption of water in a conventional way was attributed to the water that was used to wet the fields before ploughing. The second highest value for tractor ploughing was linked to deeper depths of ploughing when tractors were used. The observed least amount of water consumed by ox-plough without wetting was linked to shallow depths of ploughing because the soils were hard to plough. From the results, farmers were able to appreciate the need for not wetting the field, thus avoiding wastage of scarce water resources. The labor that would have been used during wetting the field before ploughing could be used elsewhere.

Conventional ox-ploughing resulted in a significantly lower mean yield of 4.7 tons/ha. This agrees with Huang et al. study [12] that found that deep tillage had more yield than shallow tillage. Use of the hand hoe ploughing technique increased yields by 20 percent of the conventional ox-ploughing. Yields for tractor ploughing and hand hoe ploughing were also close to 5.5 tons/ha observed by FAO (2020) [13]. All the yields were well above the average of 3.84 tons/ha [8], but they were half of the potential yields of 10 to 11 tons/ha for low-land rice when water is not limiting [24]. Furthermore, the observed yields were below 7.4 tons/ha for USA, however, the yields were closer to 6.19 tons/ha in China [24]. This shows that there is still a need for improvement in rice production in the area of study. These WUE were similar to rice production in Pakistan, which has a WUE less than 0.45 kg/m$^3$ [25].

Another way of increasing yields and water use efficiency can be done by using biochar, as was observed in China [26]. The reason why hand hoe performed better than all the other techniques could be attributed to the reason that hand levelling and harrowing was done by hand. While hand hoe had the highest WUE, it is a laborious job and time consuming. Hand hoe ploughing is impractical for larger fields, but since farmers have small pieces of land, they are recommended because they create jobs for youth and do not pollute the environment like tractors. Engine of tractors use diesel, fossil fuel, which

produces carbon dioxide when the tractor is ploughing. Carbon dioxide is one of the greenhouse gases that contribute to global warming, as observed by Mamona et al. [5]. This is welcome as the County Government of Homabay in the area has a labor force that is 48% of the population. In fact, this could assist in alleviating the rate of unemployment in the county, which stood at 73% [14]. Use of hand hoe ploughing is encouraged, as there was not much benefit in using tractor ploughing followed by subsequent harrowing and levelling manually. Thus, the study therefore encourages that hand hoe preparation techniques be used for water saving, better yields and job creation purposes. From the study, it can be seen that partial mechanization is not advantageous. The study looked at reducing the usage of irrigation water, which is a scarce resource in the world [27] and this renewable resource is expected to reduce in future [28] and not water quality. Environmental degradation in the area means high morbidity of water pollutants [14]. Since the water for irrigation comes from the river, it is suspected as having some heavy metals, as seen in Yangtze river in China [29] or emerging pollutants in drinking water resources [30], which can pose serious health risks and needs more research in the future.

## 5. Conclusions

The study investigated the effects of different ploughing techniques on the number of tillers, rooting depth, water retention, yield and water use efficiency in experimental plots under irrigation in the Maugo rice field. Use of the hand hoe ploughing technique increased yields by 20 percent of the conventional ox-ploughing. Use of water before ploughing does not add value to paddy rice production if hand ploughing is used during harrowing and levelling. Hand ploughing, harrowing and levelling resulted in the highest yield of 5.7 tons/hectare and water use efficiency of 0.49 kg/m$^3$. This research was conducted for one season and there is a need for further study to capture any climatic related conditions. Hand hoe ploughing is recommended as there were no benefits in using tractor ploughing followed by subsequent harrowing and levelling manually. Furthermore, it would be interesting to monitor, in the future, whether farmers adopted the technology and accepted it. Should the adoption be satisfactory, another combination of farmer's conventional method of ploughing and harrowing using tractor will be done.

**Author Contributions:** Conceptualization, P.K.C. and C.K.K.; methodology, J.O.; software, C.K.K.; validation, S.A.S., P.K.C. and J.O.; formal analysis, C.K.K.; investigation, P.K.C.; resources, P.K.C.; data curation, C.K.K.; writing—original draft preparation, P.K.C.; writing—review and editing, J.O.; visualization, S.A.S.; supervision, S.A.S.; project administration, J.O.; funding acquisition, S.A.S. and V.H. All authors have read and agreed to the published version of the manuscript.

**Funding:** This research received no external funding.

**Institutional Review Board Statement:** Not applicable.

**Informed Consent Statement:** Not applicable.

**Data Availability Statement:** Not applicable.

**Acknowledgments:** The authors than University of Eldoret and Ministry of Education for the study scholarship. Authors also thank farmers who allowed the research to be done in their farms.

**Conflicts of Interest:** The funders had no role in the design of the study; in the collection, analyses, or interpretation of data; in the writing of the manuscript, or in the decision to publish the results.

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
