# Peer review of "Effect of Ploughing Techniques on Water Use and Yield of Rice in Maugo Small-Holder Irrigation Scheme, Kenya"

_agriengineering, doi:10.3390/agriengineering3010007_

Round 1

Reviewer 1 Report

  • The introduction should be placed more continuously and not have so many sub-chapters.
  • In the introduction, the themes should be more connected. The same is true of the rest of the text.
  • Correction of typing errors is recommended.
  • The work should be further developed. More work should be done in particular on the discussion and conclusions 
  • The justification for pollution caused by the tractor versus manual labor should be further developed, namely at the opportunity cost of the work of small farmers. 

Author Response

Dear Reviewer,

I have gone through the comments and I have attached my responses.

Thanks

Reviewer 2 Report

The research is interesting and maybe relevant to specific geography but the manuscript needs more detailed statistical analysis and quality discussion.

Author Response

Dear Reviewer,

I have gone through the comments and I have made the changes.

Round 2

Reviewer 2 Report

Authors have made the suggested corrections.